# A Comparison of the Stability of Refined Edible Vegetable Oils under Frying Conditions: Multivariate Fingerprinting Approach

**DOI:** 10.3390/foods12030604

**Published:** 2023-02-01

**Authors:** Sandra Martín-Torres, Antonio González-Casado, Miriam Medina-García, María Soledad Medina-Vázquez, Luis Cuadros-Rodríguez

**Affiliations:** Department of Analytical Chemistry, Faculty of Sciences, University of Granada, Av. Fuentenueva s.n., E-18071 Granada, Spain

**Keywords:** refined oil, deep-frying, total polar compounds, chromatography, fingerprinting

## Abstract

The stability of highly consumed vegetable refined oils after discontinuous frying of potatoes was compared. Both those vegetable oils containing additives and those that did not were considered. Vegetable oil samples were evaluated using refractive index, anisidine and peroxide values, UV absorbance and dielectric constant-based determination of the content of total polar compounds. Chemical changes caused over the frying time were monitored and multivariate modelling of the data was carried out. A new gas chromatographic-mass spectroscopy method was intended to record a fingerprint of both polar and non-polar compound fractions. Multivariate models of chromatographic fingerprints were also developed, and the results obtained from both approaches were verified to be statistically similar. In addition, multivariate modelling also allows to differentiate among vegetable oils according to oxidation performance. Indeed, it was initially observed that olive oils presented the highest natural thermo-oxidative stability compared to other seed oils, although it should be noted that these differences were not significant when regarding olive pomace oils and seed oils containing synthetic additives.

## 1. Introduction

The chemical analysis of edible vegetable oil, which constitutes a complex multicomponent food matrix, is not an easy task. It is difficult to determine with any degree of certainty all the minority constituents in a vegetable oil matrix, due to their chemical complexity and low concentration. In addition, some of these minor constituents are only present in crude oil, and technological processing such as refining removes them [1]. A reliable and accurate determination of vegetable oils composition is essential to assess their quality and authenticity. Overall, the quality of edible oil is related to its oxidative stability, storage history, sensory characteristics, nutritional properties and culinary use [2].

Among methods of cooking, deep-frying is a highly popular method to produce palatable and desirable foods with unique characteristics of flavour, odour and colour. The frying oil acts as a homogeneous heat transfer liquid and contributes to the development of specific texture and flavour of fried foods [3,4]. Oxidative and hydrolytic deterioration of the vegetable oil occur during frying: hydrolysis of lipids, breakdown and oxidation of the unsaturated bonds of fatty chains due to the presence of oxygen, the moisture, the high oil temperature and the leaching of components from the food [5,6,7]. That results in the generation and accumulation of several degradation products, volatile or non-volatile compounds. Volatile compounds consisting of aldehydes, ketones, hydrocarbons, alcohols, acids, esters and aromatic compounds such as hexene, octane, heptadienal, decadienal and pentanol, depending on both the fatty acid composition and the state of preservation of the vegetable oil, are generated in the frying oil. The aldehydes are one of the most problematic products because they occur in highest proportions, can be absorbed into the cooked food and may have a toxic effect on human health. The volatile compounds are largely removed from the oil, lost in the atmosphere with the steam during frying and have significant implications on the flavour of both the frying oil and the fried food [8]. However, the non-volatile compounds released, collectively known as total polar compounds (TPC), remain in the frying oil [9]. They are produced primarily by thermal oxidation and polymerization reactions of unsaturated fatty acids and are of special concern due to accumulation in the frying oil, which promotes further degradation, and the fact that they are absorbed by the fried food modifying the oil’s nutritional and physiological properties [10,11]. Polar compounds mainly consist of oligomers, dimers and monomers from oxidized triacylglycerol, diacylglycerols and free fatty acids [12,13].

Standards or guidelines have been established in many countries to ensure the high quality of fried foods. The most widespread limitation states that used frying oil should be replaced when TPC reaches a defined threshold. 25% of TPC (*w*/*w*) is the legal limit value in Spain [14], Belgium, France, Portugal and Italy, while 24% is set in Germany and 27% in Australia, China and Switzerland [11]. TPC is a representative value of the new compounds formed during frying. Determination of TPC is usually performed by a previous silica gel column chromatographic fractionation followed by further gravimetric analysis, according to the IUPAC [15] and AOAC [16] standard methods. However, these methods are chemically reagent-intensive, time-consuming and require technical experts with a reliable level knowledge. The characterization of the products formed during food-frying or a simulated frying process by advanced chromatographic techniques and sophisticated methods has been extensively reported [17]. They include complex sample pre-treatments and are only used to determine the known products. Several earlier trace amounts of undetected products need to be investigated using suitable high-resolution chromatographic methods coupled to more efficient detection techniques. Moreover, some nonpolar compounds such as unchanged triacylglycerols, cyclic products and trans-isomers, also present in the frying system, are also closely related to the quality of frying vegetable oils or fried foodstuffs in respect of taste and safety [18].

Because of that, a number of attempts have been tried to find simple and rapid substitute methods. Alternatively, there are commercially available tests to determine the TPC of vegetable oils subjected to frying by monitoring the dielectric constant since an increase in the concentration of polar molecules instantly causes an increase in the value of this physico-chemical parameter in the vegetable oil matrix. The application of this method requires simple and inexpensive equipment (just an electrochemical probe) and allows for quickly addressing issues such as healthy intake estimations and safety assessments [19]. However, these devices, which need specific and careful calibration, are not suitable to be employed for all types of vegetable oils. In addition, temperature could influence the measurements and interferences such as water, salt and minerals could affect the polarity of samples and give false information about the TPC content [20]. Other alternative methods have been developed to determine the TPC content of edible oils; in this respect, FTIR is an attractive and growing analytical technique, although there are still challenges to overcome before it can be used with sufficient confidence [11].

The application of chemometric approaches in testing the quality status of frying edible vegetable oils has not been attempted widely [21]. Through this study, the time-dependent monitoring of the formation of polar and polymeric compounds and changes in physico-chemical parameters over time was performed. Discontinuous chips-batch deep-frying process (over some heating and cooling cycles) was simulated. The most commonly used edible vegetable oils, both containing synthetic antioxidants and anti-foaming agents (since they have been historically added to frying oils to extend the shelf life) and not, were used in the simulated frying processes. The main objective is to develop a non-targeted chromatographic method to obtain a characteristic instrumental ‘fingerprint’ combining the polar and non-polar fractions of the oils and to provide complete traceability of the degradation process of oils subjected to frying conditions. Gas chromatography-mass spectrometry (GC-MS) was used. In addition, to monitor changes in the physico-chemical characteristics of the oils during frying by performing classical determinations such as peroxide and anisidine values. A multivariate study of the stability and shelf life of these vegetable oils has been carried out, comparing chromatographic fingerprints behaviour and the results corresponding to classical determinations together with the use of proper chemofoodmetric tools. In addition, the equivalence between the results obtained on the frying stability of each type of vegetable oil will be statistically tested.

## 2. Materials and Methods

### 2.1. Solvents and Reagents

Cyclohexane GC grade (Fluka, Geel, Belgium), chloroform UV, IR, HPLC grade (Panreac, Barcelona, Spain), diethyl ether HPLC grade (VWR, Leuven, Belgium), n-hexane and isooctane HPLC grade (Honeywell, Muskegon, MI, USA) as solvents.

In addition, p-anisidine 99% (Alfa-Aesar, Kandel, Germany), potassium iodide 100% (VWR, Leuven, Belgium), glacial acetic acid 99.8%, sodium thiosulphate 99.5%, potassium iodate 98% and hydrochloric acid 37% (Panreac, Barcelona, Spain) were used as reagent.

Lasty, p-cymene 99% (Across, Geel, Belgium) and methyl oleate 99% (Sigma-Aldrich, Steinheim, Germany) were used standards.

### 2.2. Samples

The test vegetable oils were: 4 sunflower oils (SO), 2 olive-pomace oils (PO), 2 olive oils (marketed blends of virgin and refined olive oil) (OO) and 2 seed oils blends (consisting of high oleic sunflower, corn and soybean oils) (BO). Among them, 3 sunflower oils and 1 seed blend oil contain synthetic antioxidants and/or anti-foaming agents (vitamin C, vitamin E, ascorbyl palmitate, propyl gallate, dimethylsiloxane). All of them were purchased from local supermarkets and grocery shops; they were all qualified products.

### 2.3. Frying Experimental Procedure

During the frying experiments, 1.5 L of edible vegetable oil was placed in an electric domestic stainless clean fryer (1000 W, Cecotec, Spain) and heated up to 180 ± 10 °C (manufacturer’s recommended temperature for chips frying). When the temperature reached the stated value, 200 g of strip-cut fresh and raw potatoes (approximately 10 mm × 10 mm × 5 mm thick) previously paper-dried were placed in a basket fryer, immersed in the hot vegetable oil and fried for 5 min. During the frying process, the lid of the fryer was open (as advised in the fryer’s instructions). The chips were removed from the fryer after 5 min. The process was continued for a new potato batch for 2 h (200 g of potatoes every 30 min). After 2 h of frying the fryer is turned off and the oil is allowed to cool below 100 °C. A 10 g aliquot of vegetable oil, accurately weighted, is sampled before a further heating cycle. Immediately after cooling, each vegetable oil aliquot was packed in an amber glass vial, brought to room temperature, stored in a refrigerator and kept at 4 °C until analysis in order to avoid further chemical changes.

The potato batches were fried successively in the same vegetable oil for consecutive frying treatments until stopped when the regulatory limit of 25% TPC is exceeded or the oil content in the fryer is below the lower limit necessary to ensure that the potato strips are immersed for at least 2 cm. In summary, the frying experiment involved intermittent heating of the vegetable oil for two hours, at 180 ± 10 °C and exposed to air, without renewing the oil throughout the frying cycles.

### 2.4. Analytical Equipment

Abbe refractometer ORT1RS and Mettler Toledo G20 Compact Titrator equipped with a combined platinum electrode (Mettler Toledo DMI140-SC, Schwerzenbach, Switzerland) were used to measure refractive index and peroxide value, respectively. Molecular UV absorptivities values were measured using an Agilent 8453 spectrophotometer. An oil monitor electrochemical probe FOM320 Ebro device was employed to quickly measure total polar compound (%TPC). Technical data provided by the device supplier ensure good correlation with the recognised chromatographic method for the most common oils running in the ‘semiliquid’ equipment mode [16], and its performance was periodically checked with a suitable reference oil (Testo 05542650). An Agilent 7820A gas chromatographer equipped with an autosampler (7693) and a mass spectrometer (5977B MSD), using Rtx-65TG (30 m × 0.25 mm i.d., 0.1 μm) capillary column was applied for chromatographic analyses.

### 2.5. Physical-Chemical Analyses

Different targeted analytical parameters have been determined to conventionally evaluate the quality of frying oil which are based on the measurement of physical changes and particular chemical composition, arising as a consequence of deteriorative reactions [22,23].

#### 2.5.1. Refractive Index (RI)

Refractive index (RI) refers to the ratio of the speed of light in a vacuum to its speed through the substance concerned (the vegetable oil). The RI was determined as described by the ISO 280 standard [24]. A double prism was opened and then a few drops of vegetable oil were placed on the prism; the determination of RI was carried out using a reference temperature of 20 ± 2 °C. The refractometer was cleaned between readings by wiping off the oil with smooth tissue paper; the prism was regularly cleaned with petroleum ether and then allowed to dry by use of a clean tissue. Each measurement of RI was conducted in duplicate. Milli-Q deionised water and p-cymene were used as control standards.

#### 2.5.2. Peroxide Value (PV)

Peroxide value (PV) is the most popular index concerning the oxidation of vegetable oils. It is a direct measure of the primary oxidation taking place in vegetable oils. Oil peroxide values were determined according to the standard method described in ISO 27107 [25] by potentiometric titration. Some oils were randomly selected to be analysed in an accredited laboratory in order to verify the trueness of the results.

#### 2.5.3. Anisidine Value (AV)

Several types of oxidized triacylglycerol compounds can be produced during secondary oxidation from lipid hydroperoxides such as aldehydes, epoxides and ketones. In this respect, the anisidine value (AV) is an indicator of the concentration of aldehydes generated in heat stressed oils. AVs were determined according to the standard method described in ISO 6885 [26]. A solution of the vegetable oil in n-hexane is reacted with p-anisidine in glacial acetic acid and the UV absorbance is measured at 350 nm both before and after the reaction. The absorbance of p-anisidine solution is measured at the beginning and at the end of each batch as a reference solution of quality control.

#### 2.5.4. UV Absorption

Absorptivities at 232 nm are related to the formation of hydroperoxides (primary stage of oxidation) and conjugated dienes (intermediate stage of oxidation), while absorptivities at 268 nm are associated with the formation of carbonyl compounds (secondary stage of oxidation) and conjugated trienes (technological treatments), respectively. Specific UV absorptivities K_232_ and K_268_ were determined in compliance with COI/T.20/Doc. No 19 standard method [27]. One sample from each analysis batch was checked in an accredited laboratory to verify the reliability of the results. Both metrics have been proved to increase during the frying process [28].

#### 2.5.5. Total Polar Compounds (TPC)

The TPC monitoring of vegetable oils subjected to frying will ensure that the product retains a sufficient quality without adversely affecting the health of the consumer. Extended use of in situ rapid instrumental tests correlated with standard methods is crucial in the fast-food segment, characterized by the practice of discontinuous frying. Because of that, the %TPC was directly measured in hot vegetable oil by using the TPC meter. For each time and vegetable oil, the measurement was conducted in triplicate and the mean value was recorded. It is a direct measuring device that allows measuring the quality status of the vegetable oil through the actual dielectric constant. It incorporates a sensor that is submerged in vegetable oil and subjected to the frying process for one minute, yielding a %TPC value [29]. The electrochemical probe is usually calibrated by the supplier in such a way that a reliable measuring result can be obtained on market frying oils. In order to obtain optimum measuring results, it must be ensured to remove fried goods from the oil and wait approximately 20 min, the temperature of the oil is from 150 °C to 180 °C and the electrochemical probe is at least 2 cm away from the external wall of the vessel, hot vegetable oil with the probe should be stirred so that it will acquire the temperature of the oil as rapidly as possible and the instrument must be kept steady.

### 2.6. Chromatographic Analysis

A simple method based on GC-MS non-targeted analysis is proposed to monitor the occurrence of both polar and non-polar compounds caused by the frying process. As result, two particular GC chromatographic fingerprints are acquired for each vegetable oil which will subsequently be considered in order to reach conclusions on the shelf life.

Previously to chromatographic analyses, the non-polar and polar chemical fractions were separated by silica-solid phase extraction cartridges (SPHE-S61-030). An amount of 250 mg of the vegetable oil was diluted in 1 mL of n-hexane and then 8 mg/mL of methyl oleate was added as internal standard. The fraction containing the non-polar compounds and the internal standard (IS), was first eluted with 10 mL of n-hexane/diethyl ether (90:10, *v*/*v*). A second fraction, which includes the total polar compounds, was subsequently eluted with 10 mL of diethyl ether. After solvent evaporation, the contents (% *w*/*w*) of the non-polar fraction are determined gravimetrically and then the polar fraction is estimated by differential weighing.

After that, the polar fraction was diluted in 1 mL of chloroform while 20 mL of chloroform were needed to dilute the non-polar fraction, filtered (PTFE 0.22 μm) and both fractions were chromatographically analysed individually, using the same chromatographic method. Separation was performed on an Rtx-65TG capillary column (65% diphenyl—35% dimethylpolysiloxane; 30 m × 0.32 mm i.d. × 0.1 μm). An amount of 2 μL of diluted polar and a non-polar fraction (for each oil aliquot), split ratio 10:1, was injected at 320 °C. The oven temperature was increased from 200 °C to 290 °C in 3 min, then raised to 370 at 10 °C/min and hold for 6 min to a total run time of 17 min. MSD transfer line was set at 350 °C. Solvent delay of 1.20 min, starting m/q of 50 and ending m/q of 1000 were fixed in the detection step. The efficiency of the separation and analysis process was checked out by relative area of internal standard. The high run speed and low solvent consumption are the major advantages of this method considering both separation and chromatographic stages compared to the conventional method based on adsorption chromatography and gravimetric determination [15].

### 2.7. Chemometrics

Multivariate data evaluation was carried out using MATLAB (R2017b version, The Mathworks Inc., Natick, MA, USA) and PLS_Toolbox (Eigenvector Research Inc., Wenatchee, 193 WA, USA).

Principal component analysis (PCA) is a well-known chemometric tool to screen data and reduce the dimension of a data set. PCA is aimed at finding the simplest mathematical model able to describe the data set satisfactorily. It looks for a smaller number of underlying new variables, named principal components (PCs), which explain most of the variability exhibited by the larger number of measurements made on the objects/samples. It is an unsupervised method because it does not require training input to find the output: no additional knowledge (e.g., y-variable) besides raw data (x-variable) is needed to describe the data set. The significant PCs can be used in place of the original variables for successive treatment or to visualize the information contained in the data set. Loadings are the estimated coefficients that define the linear combination of the original variables originating from each principal component. Scores are the projections of the objects on the new PC axes [30].

Partial least-squares (PLS) regression is a multivariate linear regression technique aimed at fitting cause–effect relationships. PLS computes latent variables (LVs), linear combinations of predictors that are similar to PCs and finds the maximum correlation direction between response and LVs. PLS is a supervised multivariate method because, apart from the information on the x-variables measured (experimental data), the available knowledge on a dependent y-variable (in this study, storage time) is applied. PLS simultaneously maximizes both the LVs variance and the correlation concerning the y-variable. The complexity of the model depends on the number of significant LVs, according to the maximum percentage of cross-validated explained variance [31].

## 3. Results

As mentioned above, each oil sample was subjected to an increasing number of frying hours until exceeded the 25% TPC regulatory limit (obtained using the TPC meter) or up to a maximum of 32 h when the remaining vegetable oil result was insufficient to carry out a new frying cycle in a suitable way. The bar graph in Figure 1 shows the comparison among vegetable oils concerning both the oxidative stability and the frying time. Different behaviour is observed in the type of vegetable oil and whether or not they contain synthetic additives. In addition, the standard deviation intervals prove that all vegetable oils no having antioxidants exceeds significantly 25% TPC.

SO reaches the higher %TPC value in the shortest time, similar behaviour is shown by PO and BO oils, which would tolerate one additional frying cycle (compared with SO) without breaching the requirement. Among the vegetable oils that do not have antioxidants, OO is the one that withstands the most frying time below the regulated threshold. Seed refined oil containing synthetic additives (SO_ant and BO_ant) does not reach 25% TPC even after 32 h of frying.

Multivariate modelling. As introduced, temperature and interference substances could influence the polarity of samples and can give false information about %TPC content in the dielectric constant-based measurement of heated vegetable oil [32,33]. Multiparametric monitoring of the kinetic of oxidation and multivariate stability models have been proved to be more generically applicable than single-parameter models [21].

Slight increase in RI of frying vegetable oils (as also reported for soybean oil [34], as well as PV, which changes from 6 to 11 mEq/Kg (formerly, milliequivalents of active oxygen per kilogram of vegetable oil) for OO, were recorded. AVs highly differ between heating oils and fresh oils. UV absorbance measurements notably increased during frying finding the lowest value for olive oils and the highest one for sunflower oil, in agreement with other results [26].

Multivariate analysis of discrete experimental data was carried out. Data were arranged in a 106 × 7 matrix: each row corresponds to each oil aliquot taken (from 0 to total frying hours, respectively). Each column corresponds to the value of a single determined parameter: %TPC, RI, PV, AV, K232 and K270. The values of the experimental data are shown in the Appendix A.

Data were autoscaled (subtraction of the mean and division by standard deviation for each variable) before PCA computation to assign the same numerical weight to each variable. Four PCs were selected which explained 90% of the cumulative variance. Figure 2a shows the PC1-PC2 scores plot. PC2 scores allow grouping for vegetable oil types. The PC1 (41% of variance) seemed to group and order the samples according to frying time from negative scores for unheated oils to positive as heating time increased. If only the behaviour of PC1 scores against the sample number is considered (see Figure 2b), as frying time increases, different slopes of scores can be observed depending on the vegetable oil type and the presence/absence of synthetic additive.

Loading plot on Figure 3 shows how strongly each characteristic influences the PC1. All the original variables relate similar to the PC1, in spite of peroxide value, which exhibits lower importance.

Similar results were obtained by PLS modelling when the number of hours of frying is used as a dependent y-variable. LV1 scores (49% of the variance for y-variable) increase with frying time but this rise is not uniform and varies depending on the vegetable oil. Differences in data fitting could be also observed when a linear regression model (see Figure 4) is fitted between predicted variables and frying time (R^2^_calib_ 0.59).

Fingerprinting. Pre-processing of chromatographic fingerprints was carried out. A low-level data fusion strategy was developed: total ion current (TIC) chromatograms from polar and non-polar fractions of the same oil aliquot were concatenated on a single chromatographic fingerprint (1466 × 2 variables, total fictitious retention time of 32 min). Figure 5 shows a comparison between a polar and non-polar fussed fingerprint obtained from olive oils after 2 and 22 h of frying, respectively. It can be noted that, for the same OO, as the frying time increases, new chromatographic peaks appear in the polar fraction region. On the contrary, the peaks of the non-polar fraction are smaller in shape. These differences will be verified with the chemometric treatment of the fused matrix.

Sample fingerprints were embedded in a data matrix and subsequently pre-processed using a homemade MATLAB function, named ‘MEDINA’ (version 07) [35] (Pérez Castaño et al., 2015) and PLS toolbox pre-processing available methods. Indeed, each fingerprint was baseline-corrected using ‘Whittaker filter’ (Λ = 100, *p* = 0.001), overlaid and filtered, normalised of intensities with respect to the internal standard, aligned using ‘icoshift’ algorithm and finally mean centred. A new PCA unsupervised model was set up using the entire pre-processed fussed chromatographic fingerprints as input data set. Seven Pcs were selected explaining 73.48% of the cumulative variance. Differences were found regarding both frying time and type of vegetable oil. As an example, Figure 6 shows the PC2-PC5 scores plot in which differences are displayed. Similar results were found by PLS modelling about frying time (R^2^_cal_ 0.78).

Comparison between physico-chemical variables-based and chromatographic finger-printing-based multivariate models. In consideration of these findings, two different PLS models were set up for each vegetable oil: the recorded physico-chemical variables (discrete variables, including %TPC obtained using the electrochemical meter) and chromatographic fingerprints were modelled separately, using frying time as the input y-variable for each type of oil. Some performance metrics of both models were evaluated and the main results are shown in Table 1.

The slopes of the regression lines obtained from both methods must be compared to ensure equivalence and, therefore, that the developed GC fingerprinting method provides equal information as all other experimental measurements. Assuming that analytical signals are normally distributed, the comparison of the slopes of two regression lines could be performed using a statistic Student’s *t*-test whose null hypothesis is defined by H_0_ ≡ β1 = β2, in which β1 and β2 symbolize the slopes. More comprehensive details and full mathematical treatments are out of the scope of this work, but they can be found elsewhere [36].

Table 2 shows the statistics on the application of *t*-test formulation to the specific models. Firstly, a statistic F-test was applied to compare the squared standard errors (or residual variance, s_y/x_^2^) of the two regression lines of each example. In all cases, *p*-values were higher than 0.05 or 0.1 so the null hypothesis could not be rejected (the slopes are not significantly different) regardless of the significance level being considered.

Stability modelling. Scores of LV1 from fingerprints PLS modelling, which were time-structured, were used to establish a stability model for each vegetable oil. LV1 scores capture the largest data variance and are oriented in a way as to provide the maximum covariance between the variables for the two blocks, i.e., chromatographic fingerprints and frying time. Regression goodness of fit summary is shown on Table 3. LV1 linear fitting was proved satisfactory when plotting LV1 vs. frying time (see Figure 7).

Slopes of PLS multivariate regression models were compared to draw conclusions about the relative stability of each type of oil and thus reach a decision on the suitability of each particular vegetable oil for the discontinuous (domestic) deep-frying process. Clearly, seed oils (SO and BO) degradation occurs the fastest, showing a higher oxidation slope value than the other oils. Notice that BO containing additives show higher inertia to thermal degradation when compared to BO without additives, which, however, is similar to that shown by the PO and OO. The best performance against oxidative reactions, i.e., the lowest oxidation slope, is revealed by SO containing antioxidants.

These results differ from the initial conclusions provided by the univariate %TPC values obtained using the electrochemical meter. According to the univariate modelling based on %TPC measurements, the stability of refined edible vegetable oils which contain synthetic antioxidants and/or anti-foaming agents under frying conditions was much higher than those oils which do not. On the contrary, when multivariate modelling of the stability from the chromatographic fingerprints is applied, it appears that the BO_ant oxidation behaviour is similar to that of olive oils (and so similar to pomace oils) showing similar values of the oxidation slopes. This confirms that the model developed from the fussed polar and non-polar chromatographic fingerprints are equivalent to the multivariate modelling of all physico-chemical parameters described (discrete variables, including %TPC obtained using the electrochemical probe).

## 4. Conclusions

This study described the frying stability of most commonly consumed edible vegetable oils. A chromatographic non-targeted fingerprinting methodology has been proposed for monitoring polar and non-polar fractions caused by the frying process. The increase in the contents of total polar compounds, refractive index, peroxide value, anisidine value and UV absorptivities were found to be dependent on the frying time. Multivariate modelling of discrete measurement leads to the same conclusions as modelling of fingerprinting data. The chemometric PCA and PLS models allow differentiating among types of vegetable oils and stability against oxidation. In addition, vegetable oils containing synthetic antioxidants and/or anti-foaming agents may be distinguished from the ones that do not.

The results obtained in this study were found to be in good agreement with the literature findings. From the chromatographic fingerprint data, olive oils were found to have the highest thermo-oxidative stability compared with seed oils, such as sunflower oils or blended oils. However, this difference is not significant with regard to pomace oils and those seed oils containing synthetic additives. Nevertheless, data analysis revealed that univariate approach models, i.e., rapid measurement of %TPC using the electrochemical probe, may lead to the wrong conclusion about the stability and nutritional shelf-life of vegetable oils. On the contrary, the use of chromatographic fingerprints of both the polar and non-polar fractions provides information more reliable on the oxidation state of the oils at all time depiction allowing for more consistent healthy intake predictions based on a single, fairly straightforward analytical method.

## Figures and Tables

**Figure 1 foods-12-00604-f001:**
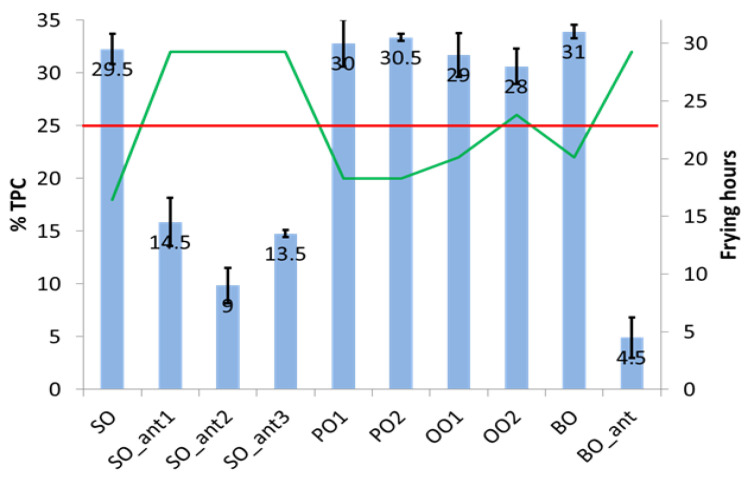
Bar chart of %TPC versus frying time (in hours, drawn by the green straight lines) of different vegetable oil samples: sunflower oil (SO), pomace oil (PO), olive oil (OO) and blended oil (BO) (“ant” is referring to synthetic antioxidants and anti-foaming). The red line is denoting the 25% TPC threshold value.

**Figure 2 foods-12-00604-f002:**
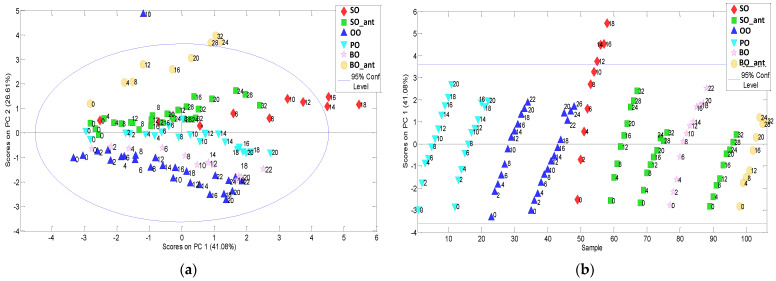
(**a**) PC2-PC1 scores plot from the targeted physico-chemical parameters modelling; (**b**) PC1 scores against the sample number plot clearly show the frying time dependence of each type of vegetable oil.

**Figure 3 foods-12-00604-f003:**
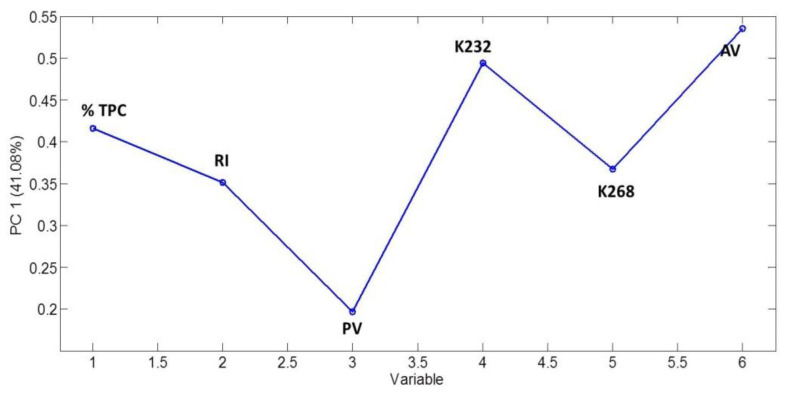
Loading plot for the PC1 of the model (TPC: total polar content; RI: refractive index; PV: peroxide value; K232: absorptivity at 322 nm; K268: absorptivity at 268; AV: anisidine value).

**Figure 4 foods-12-00604-f004:**
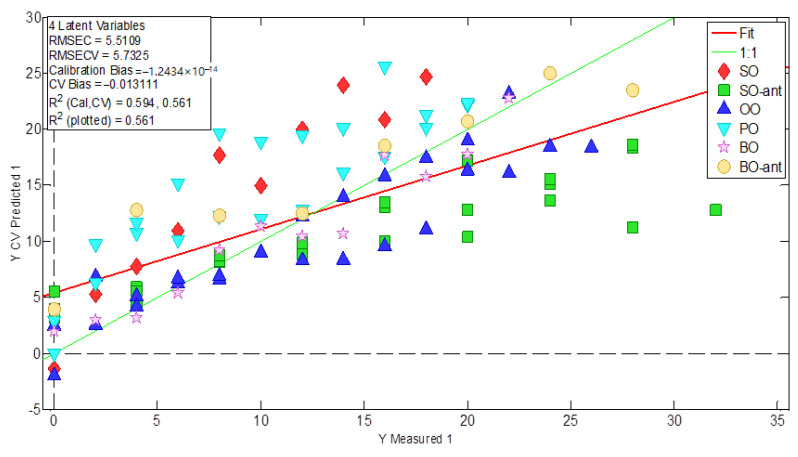
Relationship between dependent predicted discrete variables and independent variable Y (frying hours) by PLS method.

**Figure 5 foods-12-00604-f005:**
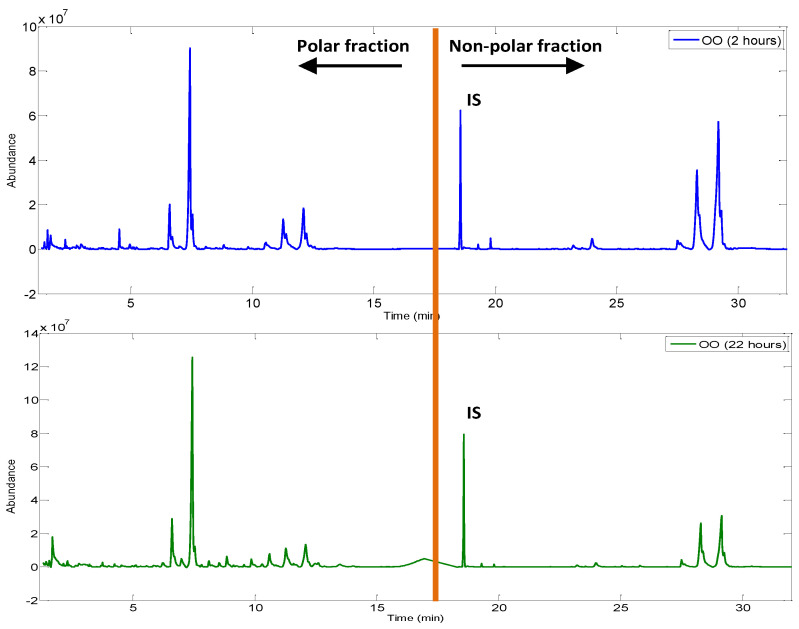
Fussed fingerprints of olive oil (OO) after two and twenty-two hours of frying, respectively (IS symbolizes the internal standard).

**Figure 6 foods-12-00604-f006:**
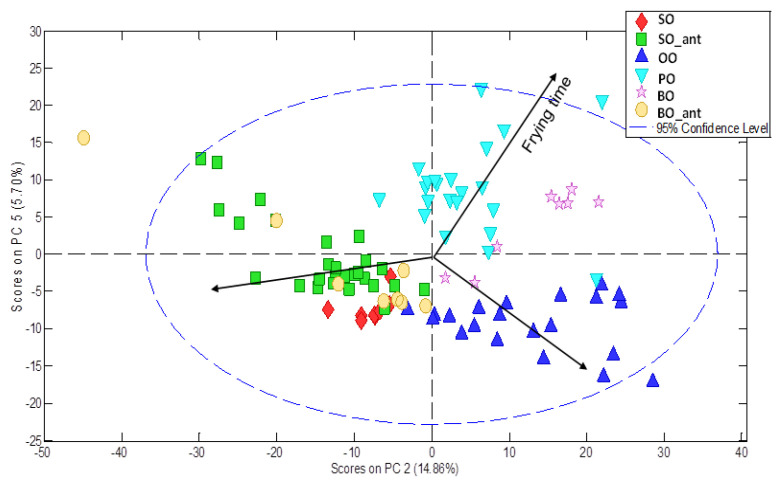
PC2-PC5 scores plot from fussed chromatographic fingerprinting data modelling.

**Figure 7 foods-12-00604-f007:**
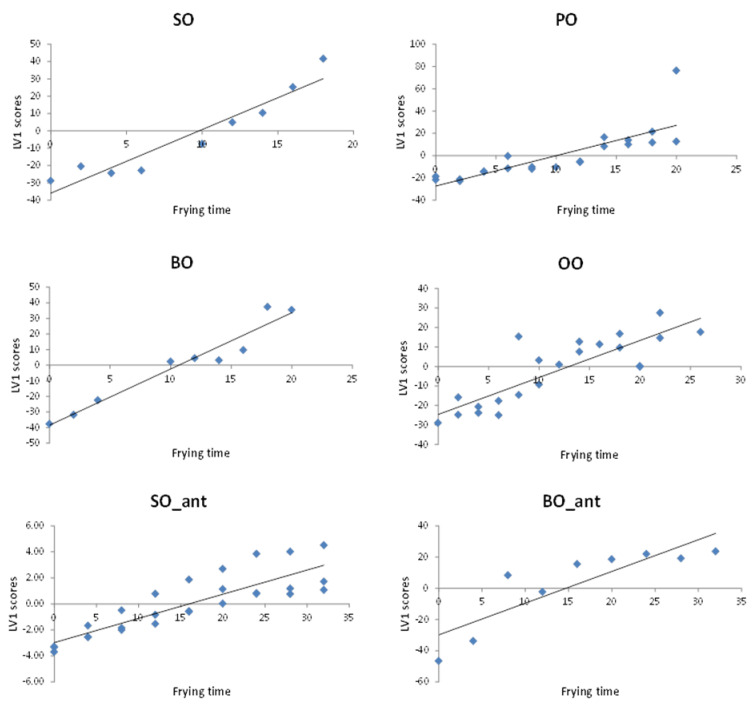
LV1 scores plotting vs. time of frying. SO is sunflower oil; PO is pomace oil; BO is a blend of different seed oil; OO is olive oil and _ant symbolises an oil containing additives.

**Table 1 foods-12-00604-t001:** Performance metrics of PLS model carried out for each vegetable oil.

	Fingerprinting Modelling			Discrete Variables Modelling		
Vegetable Oil	PLS Model	RMSECV	Y CV Predicted vs. Frying Time	PLS Model	RMSECV	Y CV Predicted vs. Frying Time
Sunflower oil (SO)	3 LV 73.2% X-var 99.4% Y-var	3.2145	y = 1.0553 x + 0.5364 R² = 0.817	1 LV 80.1% X-var 95.8% Y-var	1.3383	y = 0.9583 x + 0.3689 R² = 0.951
Pomace oil (PO)	3 LV, 65.3% X-var 99.5% Y-var	4.2901	y = 0.9621 x + 0.8200 R² = 0.681	2 LV 73.6% X-var 98.2% Y-var	1.3256	y = 0.9891 x − 0.0552 R² = 0.960
Blended oil (BO)	2 LV 62.2% X-var 97.2% Y-var	2.9666	y = 0.8845 x + 1.337 R² = 0.945	2 LV 78.9% X-var 98.9% Y-var	2.7933	y = 0.8518 x + 0.7544 R² = 0.872
Olive oil (OO)	2LV 48.2% X-var 88.1% Y-var	3.0873	y = 0.7670 x + 3.1144 R² = 0.833	4 LV 98.6% X-var 68.1% Y-var	6.8781	y = 0.6653 x + 2.5972 R² = 0.378
Sunflower oil containing additives (SO_ant)	5 LV 73.8% X-var 98.9% Y-var	4.5728	y = 0.8245 x + 3.1474 R² = 0.775	2 LV 81.9% X-var 89.6% Y-var	4.2074	y = 0.8256 x − 2.9714 R² = 0.741
Blended oil- containing additives (BO_ant)	3 LV 8.8% X-var 97.3% Y-var	3.8987	y = 0.9804 x − 0.0605 R² = 0.814	2 LV 90.1% X-var 97.2% Y-var	3.7846	y = 1.0238 x − 1.2166 R² = 0.892

**Table 2 foods-12-00604-t002:** Statistical comparison of results obtained by fingerprinting modelling and discrete variable modelling.

Vegetable Oil	Model	Slope ± s_b_	S_y/x_	*p*-Value
Sunflower oil (SO)	Fingerprinting	1.055 ± 0.189	3.42	0.645
	Discrete variables	0.985 ± 0.082	1.49	
Pomace oil (PO)	Fingerprinting	0.962 ± 0.150	3.84	0.865
	Discrete variables	0.989 ± 0.046	1.37	
Blended oil (BO)	Fingerprinting	0.884 ± 0.081	1.65	0.827
	Discrete variables	0.851 ± 0.115	2.66	
Olive oil (OO)	Fingerprinting	0.767 ± 0.076	3.16	0.619
	Discrete variables	0.665 ± 0.177	6.55	
Sunflower oil containing additives (SO_ant)	Fingerprinting	0.824 ± 0.088	4.77	0.993
	Discrete variables	0.825 ± 0.073	3.94	
Blended oil containing additives (SO_ant)	Fingerprinting	0.980 ± 0.177	5.49	0.848
	Discrete variables	1.023 ± 0.134	4.17	

S_b_ = standard deviation of the slope; S_y/x_ = residual standard deviation; *p*-value: calculated following the *t*-statistic Welch’s test [34] when the null hypothesis of the F-test can be rejected.

**Table 3 foods-12-00604-t003:** The goodness of fitting statistics for stability models of edible vegetable oil.

Vegetable Oil	Regression Model	R^2^	RMSE
Sunflower oil (SO)	t = 3.665·LV1 − 35.89	0.9082	7.989
Blended oil (BO)	t = 3.600·LV1 − 38.40	0.9443	6.744
Pomace oil (PO)	t = 2.053·LV1 − 23.48	0.8835	7.818
Olive oil (OO)	t = 1.996·LV1 − 27.17	0.8089	7.526
Sunflower oil containing additives (SO_ant)	t = 0.815·LV1 − 19.97	0.6297	16.11
Blended oil-containing additives (BO_ant)	t = 2.171·LV1 − 33.91	0.8419	11.02

t = frying hours; LV1 = scores of the latent variable 1 from PLS model; R^2^ = coefficient of determination; RMSE = root mean squared error.

## Data Availability

Data is contained within the article or Appendix A.

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
