# Peer review of "A Comparison of the Stability of Refined Edible Vegetable Oils under Frying Conditions: Multivariate Fingerprinting Approach"

_foods, 2023, doi:10.3390/foods12030604_

Round 1
Reviewer 1 Report
Today, frying is the most common method of heat treatment of food and always causes doubts about the nutritional and health aspects. That is why the idea of this work is very current, and the results are always interesting to the scientific public.
The introduction, as well as the еxperiment design, are acceptable. However, there is one doubt about samples. Namely, a 10 g aliquot of vegetable oil was taken from each batch. Was this amount of oil enough to carry out all the experimental methods and repeat them three times.
Also, in what quantity and combination were the antioxidants added to the oil samples? Is it the same for all samples with antioxidants (SO_ant1, SO_ant2, SO_ant3 and BO_ant)? It is not specified and can significantly affect the results.
The presented methods are very precise and very important from a scientific point of view. However, rapid measurement of %TPC using the electrochemical probe is good enough, more acceptable in practice, and can be relied on.
Author Response
Reviewer #1:
Today, frying is the most common method of heat treatment of food and always causes doubts about the nutritional and health aspects. That is why the idea of this work is very current, and the results are always interesting to the scientific public.
We thank to the reviewer for his/her good first impression about the opportunity and relevance of this paper and for his/her recommendations. In the following lines the changes suggested by the reviewer are properly considered.
The introduction, as well as the еxperiment design, are acceptable. However, there is one doubt about samples. Namely, a 10 g aliquot of vegetable oil was taken from each batch. Was this amount of oil enough to carry out all the experimental methods and repeat them three times.
Only refractive index and total polar compounds measurements have been repeated for several times. Instead of that, other strategies of quality control have been used in each determination as stated on manuscript. Because of that, 10 g aliquot was enough. A larger sample amount would not have allowed a sufficient number of frying cycles/hours to be achieved.
Also, in what quantity and combination were the antioxidants added to the oil samples? Is it the same for all samples with antioxidants (SO_ant1, SO_ant2, SO_ant3 and BO_ant)? It is not specified and can significantly affect the results.
The vegetable oils containing antioxidants are marketed products and were purchased directly; no antioxidants were added in the laboratory. The additives stated on the labelling were:
â–ª SO_ant1: Vitamin E, ascorbyl palmitate, propyl gallate, and E900 agent.
â–ª SO_ant2: E900
â–ª SO_ant3: antioxidants E306 and E304 and E900 agent.
â–ª BO_ant: E900
The amounts of each additive were not indicated in any case. As the additive content was considered as a non-influential variable for forced oxidation modelling, no further information has been specified in the manuscript as it was considered to be unnecessary.
The presented methods are very precise and very important from a scientific point of view. However, rapid measurement of %TPC using the electrochemical probe is good enough, more acceptable in practice, and can be relied on.
We fully agree with the reviewer. However, to be consistent, its suitability and performance were tested in this study.

Reviewer 2 Report
The article entitled " A comparison of the stability of refined edible vegetable oils under frying conditions: multivariate fingerprinting approach" deals with interesting topics concerning the quality control of vegetable oils. In my opinion, the quality of the article is reduced by the lack of reference to specific experimental data
Some remarks:
Authors should add specific data from the analyzes of such parameters as: RI, PV, AV, K232 and K270 for tested vegetable oils.
The Authors did not provide a PCA loading plot.
p. 6 lines 276-277 Is the following statement based on statistical analysis
“In addition, the standard deviation intervals prove that all vegetable oils no having antioxidants exceeds significantly 25% TPC.”
p.7 line 285-287 Does Figure 1 shows exceeding the regulated threshold for the OO sample after 20 hours of frying
Author Response
Reviewer #2:
The article entitled "A comparison of the stability of refined edible vegetable oils under frying conditions: multivariate fingerprinting approach" deals with interesting topics concerning the quality control of vegetable oils. In my opinion, the quality of the article is reduced by the lack of reference to specific experimental data.
The authors want to express thanks for the positive feedback and appreciate the helpful comments to improve this review quality. In the following lines, we address all the issues suggested by reviewer.
Authors should add specific data from the analyzes of such parameters as: RI, PV, AV, K232 and K270 for tested vegetable oils.
A table showing the experimental results has been included in the supplementary material.
The Authors did not provide a PCA loading plot.
PCA loading plot has been added on manuscript (Figure 3).
- 6 lines 276-277 Is the following statement based on statistical analysis: “In addition, the standard deviation intervals prove that all vegetable oils no having antioxidants exceeds significantly 25% TPC.”
Indeed, the authors intended to emphasise this fact, which can be observed in the bar graph.
p.7 line 285-287 Does Figure 1 shows exceeding the regulated threshold for the OO sample after 20 hours of frying
In this case, we were referring to the mean value of the two different samples considered. However, to avoid confusion, this statement has been removed.